# You Only Transfer What You Share: Intersection-Induced Graph Transfer Learning for Link Prediction

**Wenqing Zheng**[*]                                                   *w.zheng@utexas.edu*
*The University of Texas at Austin, Austin, TX, USA*

**Edward W Huang**                                                   *ewhuang@amazon.com*
*Amazon, Palo Alto, CA, USA*

**Nikhil Rao** [†]                                                   *nikhilrao@microsoft.com*
*Microsoft*

**Zhangyang Wang** [‡]                                                   *atlaswang@utexas.edu*
*The University of Texas at Austin, Austin, TX, USA*

**Karthik Subbian**                                                   *ksubbian@amazon.com*
*Amazon, Palo Alto, CA, USA*

**Reviewed on OpenReview:** *https://openreview.net/forum?id=Nn71AdKyYH*

## Abstract

Link prediction is central to many real-world applications, but its performance may be hampered when the graph of interest is sparse. To alleviate issues caused by sparsity, we investigate a previously overlooked phenomenon: in many cases, a densely connected, complementary graph can be found for the original graph. The denser graph may share nodes with the original graph, which offers a natural bridge for transferring selective, meaningful knowledge. We identify this setting as *Graph Intersection-induced Transfer Learning* (GITL), which is motivated by practical applications in e-commerce or academic co-authorship predictions. We develop a framework to effectively leverage the *structural prior* in this setting. We first create an intersection subgraph using the shared nodes between the two graphs, then transfer knowledge from the source-enriched intersection subgraph to the full target graph. In the second step, we consider two approaches: a modified label propagation, and a multi-layer perceptron (MLP) model in a teacher-student regime. Experimental results on proprietary e-commerce datasets and open-source citation graphs show that the proposed workflow outperforms existing transfer learning baselines that do not explicitly utilize the intersection structure.

## 1 Introduction

Link prediction (Lichtenwalter et al., 2010; Zhang & Chen, 2018; Safdari et al., 2022; Yang et al., 2022b; Nasiri et al., 2022) is an important technique used in various applications concerning complex network systems, such as e-commerce item recommendation (Chen et al., 2005), social network analysis (Al Hasan & Zaki, 2011), knowledge graph relation completion (Kazemi & Poole, 2018), and more. State-of-the-art methods leverage Graph Neural Networks (GNNs) to discover latent links in the system. Methods such as Graph

---

[*]Work done during internship in Amazon.
[†]Work done while working in Amazon.
[‡]Work done while visiting Amazon.

AutoEncoder (GAE) (Kipf & Welling, 2016), SEAL (Zhang & Chen, 2018; Zhang et al., 2021a), PLNLP (Wang et al., 2021), and Neo-GNN (Yun et al., 2021) perform reliably on link prediction when the target graph has a high ratio of edge connectivity. However, many real-world data is *sparse*, and these methods are less effective in these situations. This issue is known as the "cold-start problem," and has recently been studied in the context of e-commerce (Zheng et al., 2021) and social networks (Leroy et al., 2010).

One solution to address the difficulties related to cold-start settings is transfer learning (Gritsenko et al., 2021; Cai et al., 2021). To alleviate sparsity issues of the target graph, transfer learning seeks to bring knowledge from a related graph, i.e., the *source graph*, which shares similar structures or features with the target graph. The source graph should have better observable connectivity. If such a related source graph can be found, its richer connectivity can support the target graph, augment its training data, and enhance latent link discovery.

However, transferring knowledge between graphs poses significant challenges. These challenges are (Kan et al., 2021; Zhu et al., 2021a), mainly due to differences in optimization objectives and data distribution between pre-training and downstream tasks (graphs) (Han et al., 2021; Zhu et al., 2021a). To this end, Ruiz et al. (2020) theoretically bounded the transfer error between two graphs from the same "graphon family," but this highly restrictive assumption limits its applications in real-world scenarios. Another series of studies have examined the transferability of generally pre-trained GNNs (Hu et al., 2020a; You et al., 2020b; Hu et al., 2020b), aiming to leverage the abundant self-supervised data for auxlary tasks (Hwang et al., 2020). However, they do not study which self-supervised data or tasks are more beneficial for downstream tasks. Other GNN transfer learning methods leverage a meta-learning framework (Lan et al., 2020) or introduce domain-adaptive modules with specified losses (Wu et al., 2020), but they fail to capture the potential structural prior when the source and target graphs have shared nodes.

To better exploit the potential of source-target graph transfer, we observe a widespread structural prior: the source graph may share an *intersection* subgraph with the sparse target graph, i.e., they may have nodes and edges in common. Next, we first discuss a few real-world examples before further specifying this setting.

## 1.1 Motivating Examples

Our setting assumes that given one graph of interest, we can find another graph with a common subset of nodes. Furthermore, the second graph has richer link information than the original graph. We refer to the second graph as the source graph, which we employ to transfer knowledge. We refer to the original graph as the target graph. We conceptually illustrate this setting in Figure 1. This setting is motivated by a few important real-world examples, and we discuss two of them drawn from **global e-commerce** and **social network** scenarios.

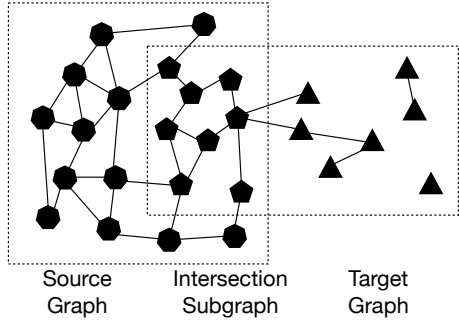

Source Graph    Intersection Subgraph    Target Graph

In global e-commerce stores such as Amazon, eBay, or Taobao, the product items and customer queries constitute bipartite graphs. The products and user queries are defined as nodes, with the user behaviors (clicks, add-to-carts, purchases, etc.) defined as edges. These graphs can be *huge in size*, and are instrumental for customizing query recommendations, predicting search trends, and improving the search experience. To improve the recommendation engine, one may formulate the information filtering process as a link prediction task and then train GNN models to predict user behavior data.

Figure 1: An illustration of the proposed GITL setting. In this setting, the target graph is sparse, while the source graph has rich link information. The source graph and the target graph are assumed to have shared nodes and edges. The goal is to use the rich information in the source graph to improve link prediction in the target graph by exploiting the structural prior.

These global e-commerce stores operate in multiple locales simultaneously. Among the emerging (smaller) locales, one practical and critical challenge commonly arises: these locales have not received rich enough user behavior, which may lead to the *cold-start* issue (Hu et al., 2021; Zhang et al., 2021c; Zheng et al., 2021). In other words, very few customer interactions result in a sparse and noisy graph, leading to less reliable predictions.

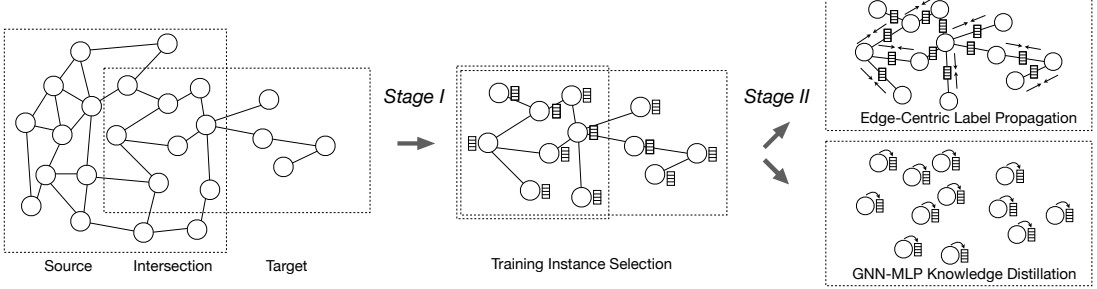

Figure 2: A visualization of the proposed framework, which contains two stages. The first stage identifies the training dataset for the model, and the second stage investigates two broadcasting approaches: the modified edge-centric label propagation, and the knowledge distillation MLP.

To improve prediction performance in these emerging locales, we can leverage rich behavioral data from more established locales, which may have years of user activity. In this case, one structural prior facilitates such a transfer: many items are available in multiple locales, and some query words might also be used by customers in multiple locales. These *shared nodes* (products and user queries) naturally bridges the two graphs. Note that the emerging and established locale graphs may have different node feature distributions. This *domain gap* arises from differences in the items available across locales, as well as the customer behavior differences related to societal, economic, cultural, or other reasons.

Other examples can be found in social networks. For instance, academic collaborations can be modeled as a graph where the nodes are authors and the edges indicate collaborations or co-authored papers. One task in such a graph is to predict co-authorship links. In this task, we can once again formulate the source-target transfer: the source graph can be an established field where authors collaborate extensively, and the target graph can be an emerging field with fewer collaborations. As another formulation, the source graph can be the author collaborations in past years and the target graph can refer to projected collaborations in future years. In both formulations, we can identify a shared subgraph: the shared nodes are the common authors, and the shared edges are pairs of authors who have publications in both disciplines (or within the same year).

With these real-world examples, we summarize the common properties of these tasks, and formulate them into a new setting, which we term as the *Graph Intersection-induced Transfer Learning* (**GITL**). In a nutshell, the GITL setting represents the cases where the source graph shares nodes with the target graph, so that we can broadcast the source graph's richer link information to the target graph via the common subgraph.

## 1.2 Contributions

We propose a framework that tackles the GITL setting from two angles: *training instance optimization* and *prediction broadcast*. Our framework addresses these two aspects using a two-stage learning process, as shown in Figure 2. For training instance optimization, we leverage the shared subset of nodes as the key structural information to transfer from the source graph to the target graph. In this step, the GNN model is trained only on the shared subgraph instead of the full source graph, which we show through experiments as a more effective method of transfer. For the prediction broadcast, we design a novel label propagation approach, which shifts the node-based graph to an edge-centric graph. This avoids over-smoothing during the broadcast. We also study a pointwise MLP model via teacher-student knowledge distillation.

Our method falls into the category of *instance-level transfer* (Pan & Yang, 2009; Koh & Liang, 2017; Wang et al., 2018; 2019). Distinct from other transfer learning approaches that fine-tune pre-trained GNNs on the target domain (dubbed the *parameter-level transfer* (Pan & Yang, 2009)), the instance-level transfer *selects* or *re-weights* the samples from the source domain to form a new training set using guidance from the target domain. As a result, models trained on these *processed* source domain samples generalize better without fine-tuning the model weights. Our method is an instantiation of this approach, as we first leverage the structure overlap prior to selecting the training instances, then re-weight the sample predictions via

source-to-target broadcast. We consider this instance-level transfer better suited for our setting, as the graph model is lightweight and easy to optimize. On the other hand, since the training data is usually massive, dropping samples is unlikely to cause model underfitting. Our contributions are outlined as follows:

- We formulate GITL, a practically important graph transfer learning setting that exists in several real-world applications. We propose a novel framework to optimize the GITL setting, which leverages the intersection subgraph as the key to transfer important graph structure information.

- The proposed framework first identifies the shared intersection subgraph as the training set, then broadcasts link information from this subgraph to the full target graph. We investigate two broadcasting strategies: a label propagation approach and a pointwise MLP model.

- We show through comprehensive experiments on proprietary e-commerce graphs and open-source academic graphs that our approach outperforms other state-of-the-art methods.

### 1.3 Related works

**Link Prediction in Sparse Graphs.** The link prediction problem is well studied in literature (Liben-Nowell & Kleinberg, 2007), with many performant models (Singh et al., 2021; Wang et al., 2021; Yun et al., 2021; Zhang et al., 2021a; Zhang & Chen, 2018; Subbian et al., 2015; Zhu et al., 2021b) and heuristics (Chowdhury, 2010; Zhou et al., 2009; Adamic & Adar, 2003; Newman, 2001). Unfortunately, most methods are hampered by link sparsity (i.e., low edge connection density). Mitigating the link prediction challenge in sparsely connected graphs has attracted considerable effort. Some suggest that improved node embeddings can alleviate this problem (Chen et al., 2021a), e.g., using similarity-score-based linking (Liben-Nowell & Kleinberg, 2007) or auxiliary information (Leroy et al., 2010). However, these methods do not yet fully resolve sparsity challenges. Bose et al. (2019); Yang et al. (2022a) treated few-shot prediction as a meta-learning problem, but their solutions depend on having many sparse graph samples coming from the same underlying distribution.

**Graph Transfer Learning.** While transfer learning has received extensive research in deep learning research, it remains highly non-trivial to transfer learned structural information across different graphs. Gritsenko et al. (2021) theoretically showed that a classifier trained on embeddings of one graph is generally no better than random guessing when applied to embeddings of another graph. This is because general graph embeddings capture only the relative (instead of absolute) node locations.

GNNs have outperformed traditional approaches in numerous graph-based tasks (Sun et al., 2020b; Chen et al., 2021b; Duan et al.). While many modern GNNs are trained in (semi-)supervised and dataset-specific ways, recent successes of self-supervised graph learning (Veličković et al., 2019; You et al., 2020a; Sun et al., 2020a; Lan et al., 2020; Hu et al., 2020a; You et al., 2020b; Hu et al., 2020b) have invoked the interest in transferring learned graphs representations to other graphs. However, transferring them to node/link prediction over a different graph has seen limited success, and is mostly restricted to graphs that are substantially similar (You et al., 2020b; Hu et al., 2020b). An early study on graph transfer learning under shared nodes (Jiang et al., 2015) uses "common nodes" as the bridge to transfer, without using graph neural networks (GNN) or leveraging a selected subgraph. Wu et al. (2020) addressed a more general graph domain adaptation problem, but only when the source and target tasks are homogeneous. Furthermore, most GNN transfer works lack a rigorous analysis on their representation transferability, save for a few pioneering works (Ruiz et al., 2020; Zhu et al., 2021a) that rely on strong similarity assumptions between the source and target graphs.

Also related to our approach is *entity alignment* across different knowledge graphs (KGs), which aims to match entities from different KGs that represent the same real-world entities (Zhu et al., 2020; Sun et al., 2020b). Since most KGs are sparse (Zhang et al., 2021b), entity alignment will also enable the enrichment of a KG from a complementary one, hence improving its quality and coverage. However, GITL has a different focus, as we assume the shared nodes to be already known or easily identifiable. For example, in an e-commerce network, products have unique IDs, so nodes can be easily matched.

**Instance-Level Transfer Learning.** Though model-based transfer has become the most frequently used method in deep transfer learning, several research works have shown the significance of data on the effectiveness of deep transfer learning. Koh & Liang (2017) first studied the influence of the training data on the testing

loss. They provide a Heissian approximation for the influence caused by a single sample in the training data. Inspired by this work, Wang et al. (2018) proposed a scheme centered on data dropout to optimize training data. The data dropout approach loops for each instance in the training set, estimates its influence on the validation loss, and drops all "bad samples." Following this line, Wang et al. (2019) proposes an instance-based approach to improve deep transfer learning in a target domain. It first pre-trains a model in the source domain, then leverages this model to optimize the training data of the target domain by removing the training samples that will lower the performance of the pre-trained model. Finally, it fine-tunes the model using the optimized training data. Though such instance-wise training data dropout optimization does yield improved performance, it requires a time complexity of $\mathcal{O}(N_{\text{training\_set\_size}} * N_{\text{validation\_set\_size}})$, which can be prohibitively costly in large graphs or dynamic real-world graphs that are constantly growing.

**Label Propagation.** Label propagation (LP) (Zhu, 2005; Wang & Zhang, 2007; Karasuyama & Mamitsuka, 2013; Gong et al., 2016; Liu et al., 2018) is a classical family of graph algorithms for semi-supervised transductive learning, which diffuses labels in the graph and makes predictions based on the diffused labels. Early works include several semi-supervised learning algorithms such as the spectral graph transducer (Joachims, 2003), Gaussian random field models (Zhu et al., 2003), and label spreading (Zhou et al., 2004). Later, LP techniques have been used for learning on relational graph data (Koutra et al., 2011; Chin et al., 2019). More recent works provided theoretical analysis (Wang & Leskovec, 2020) and also found that combining label propagation (which ignores node features) with simple MLP models (which ignores graph structure) achieves surprisingly high node classification performance (Huang et al., 2020).

However, LP is not suitable for the link prediction task. LP is prone to over-smoothing (Wang & Leskovec, 2020) for nodes within the same class, which may also hurt link prediction. For these reasons, we focus on an edge-centric variation of LP, discussed in Section 2.3.

## 2 Model Formulations

### 2.1 Notations And Assumptions

Denote the source graph as $\mathcal{G}^{\text{src}}$ and the target graph as $\mathcal{G}^{\text{tar}}$, and denote their node sets as $\mathcal{V}^{\text{src}}$ and $\mathcal{V}^{\text{tar}}$, respectively. We make the following assumptions:

1. **Large size**: the source and the target graphs are large, and there are possibly new nodes being added over time. This demands simplicity and efficiency in the learning pipeline.

2. **Distribution shift**: node features may follow different distributions between these two graphs. The source graph also has relatively richer links (e.g., higher average node degrees) than the target graph.

3. **Overlap**: there are common nodes between the two graphs, i.e., $\mathcal{V}^{\text{src}} \bigcap \mathcal{V}^{\text{tar}} \neq \emptyset$. We frame the shared nodes as a bridge that enables effective cross-graph transfer. We do not make assumptions about the size or ratio of the common nodes.

In the following sections, we describe the details of the proposed framework under the GITL setting, which consists of two stages.

### 2.2 GITL Stage I: Instance Selection

We consider the instance-level transfer, which selects and/or re-weights the training samples to improve the transfer learning performance in the target domain. Previous instance-level transfer research leverages brute-force search (Wang et al., 2019): it loops for all instances and drops the ones that negatively affect performance. However, most real-world graphs are large in size and are constantly growing (*Assumption 1*), which makes such an approach infeasible. We instead leverage a practically simple instance selection method, choosing the *intersection subgraph* between the source and target graphs as the training data. While this seems to be counter-intuitive to the "more the better" philosophy when it comes to transferring knowledge, we find it more effective in Section 3, and refer to it as the *negative transfer* phenomenon.

Specifically, we compare three different learning regimes. ❶ *target → target* (*Tar. → Tar.*) directly trains on the target graph without leveraging any of the source graph information. ❷ *union → target* (*Uni. →*

*Tar.*), where the training set is the set of all source graph edges, plus part of the available edges in the target graphs. The dataset size is the biggest among the three regimes, but no instance optimization trick is applied and it is therefore suboptimal due to the observed *negative transfer* phenomenon. ❸ *intersection → target* (*Int. → Tar.*) is our proposed framework, which trains on the intersection subgraph. The training nodes are selected based on the intersection structural prior, and the edges are adaptively enriched based on the source graph information. Specifically, we first extract the common nodes in the two graphs $\mathcal{G}^{\text{src}}$ and $\mathcal{G}^{\text{tar}}$:

$$\mathcal{V}^* = \mathcal{V}^{\text{src}} \bigcap \mathcal{V}^{\text{tar}} \tag{1}$$

Next, all edges from the source and target graphs that have one end node in $\mathcal{V}^*$ are united together to form the intersection graph, i.e., the intersection graph is composed of common nodes $\mathcal{V}^*$ and edges from either of the two graphs. Then, we build a positive edge set $\mathcal{E}^+ = \mathcal{E}^*$ and sample a negative edge set $\mathcal{E}^-$.

**Train/Test Split.** Under the graph transfer learning setting, we care about the knowledge transfer quality in the target graph. Therefore, the most important edges are those with at least one node exclusive to the target graph. To this end, the positive and negative edges with both end nodes in the source graph are used as the training set. 20% of positive and negative edges that have at least one node outside the source graph are also added to the training set. Finally, the remaining positive and negative edges are evenly split into the validation set and testing set.

### 2.3 GITL Stage II: Source-to-Target Broadcast

In stage II, we first train a GNN model for the link prediction task to generate initial predictions, then use label propagation or MLP-based methods to broadcast the predictions to the entire target graph.

**Training the GNN Base Model for Link Prediction.** Our GNN model training follows the common practice of link prediction benchmarks (Wang et al., 2021; Zhang et al., 2021a; Zhang & Chen, 2018). For the construction of node features, we concatenate the original node features $\mathbf{X}$ (non-trainable) with a randomly initialized trainable vector $\mathbf{X}'$ for all nodes. On the output side, the GNN generates $d$-dimensional node embeddings for all nodes via $\mathbf{Y} = \text{GNN}(\mathbf{A}, [\mathbf{X}, \mathbf{X}'])$. For node pair $(i, j)$ with their embeddings $\mathbf{Y}[i] \in \mathbb{R}^d$ and $\mathbf{Y}[j] \in \mathbb{R}^d$, the link existence is predicted as the inner product:

$$z_{i,j} = <\mathbf{Y}[i], \mathbf{Y}[j]> \tag{2}$$

where $z_{i,j} > 0$ indicates a positive estimate of the link existence and vice versa. We randomly sample positive edge set $\mathcal{E}^+$ and negative edge set $\mathcal{E}^-$ to train the GNN model. The model is trained with the common AUC link prediction loss (Wang et al., 2021):

$$\min_{\Theta, \mathbf{X}'} \sum_{e^+ \in \mathcal{E}^+, e^- \in \mathcal{E}^-} (1 - \mathbf{Z}[e^+] + \mathbf{Z}[e^-])^2 \tag{3}$$

After the model is trained, we have the predictions $z_{i,j}$ for all edges between node $i$ and $j$. These predictions are already available for downstream tasks. However, in the proposed framework, we further leverage label propagation as a post-processing step, which broadcasts the information from the source graph to the target graph. Specifically, we concatenate $z_{i,j}$ for all edges $e = (i, j)$ in $\mathcal{E}^+$ and $\mathcal{E}^-$, and get $\mathbf{Z} \in \mathbb{R}^{(|\mathcal{E}^+| + |\mathcal{E}^-|) \times 1}$.

**Broadcasting Predictions with Edge Centric Label Propagation.** Label propagation (LP) is simple to implement and is hardware friendly, easy to parallelize on GPUs, and also fast on CPUs. Generic LP methods (Zhu, 2005; Huang et al., 2020) diffuse the node embeddings across edges in the graph. To avoid the *over-smoothness* induced by the node-level diffusion of traditional LP, we shift the role of nodes into edges, and propose an edge-centric variant of LP. We term this method as logit diffusion-based LP (Logit-LP).

Denote $N$ as the number of nodes in the graph to be broadcast. The LP requires two sets of inputs: the stack of the *initial embedding* of all nodes, denoted as $\mathbf{Z}^{(0)} \in \mathbb{R}^{N \times d}$, and the *diffusion source embedding*, denoted as $\mathbf{G} \in \mathbb{R}^{N \times d}$. The diffusion procedure generally used in LP methods (Zhu, 2005; Wang & Zhang, 2007; Karasuyama & Mamitsuka, 2013; Gong et al., 2016; Liu et al., 2018) can be summarized into the formula below, which iterates $k$ until it reaches a predefined maximum value $K_{\max}$:

$$\mathbf{Z}^{(k+1)} = \alpha \mathbf{A} \mathbf{Z}^{(k)} + (1 - \alpha)\mathbf{G} \tag{4}$$

In our method, the Logit-LP uses an *edge-centric* view to model the diffusion procedure. As visualized in Figure 3, it shifts the role of edges into nodes, i.e., it builds a new graph $\tilde{\mathcal{G}}$, which consists of the original edges as its nodes. The idea is that it operates on an edge-centric graph $\tilde{\mathcal{G}}$ with "switched roles," where the nodes in $\tilde{\mathcal{G}}$ are edges in $\mathcal{G}$, and the edges in $\tilde{\mathcal{G}}$ represent the connectivity of edges in $\mathcal{G}$: if two edges in $\mathcal{G}$ share a node, then there is a corresponding edge in $\tilde{\mathcal{G}}$. It shifts the embeddings from the node embeddings to the edge embeddings. Mathematically, this is done by re-building the adjacency matrix $\tilde{\mathbf{A}}$, the initial embedding $\tilde{\mathbf{Z}}^{(0)}$, and the diffusion source embedding $\tilde{\mathbf{G}}$. The new nodes are binary labeled: the positive and negative edges in the original graph. The number of new nodes is $|\tilde{\mathcal{G}}| = |\mathcal{E}^+| + |\mathcal{E}^-|$. The embeddings of these nodes, denoted as $\tilde{\mathbf{Z}}_{\text{Logit-LP}}$, are edge prediction logits. In other words, the initial embedding is inherited from the GNN: $\tilde{\mathbf{Z}}_{\text{Logit-LP}}^{(0)} = \text{vec}(z_{i,j}) \in \mathbb{R}^{(|\mathcal{E}^+|+|\mathcal{E}^-|) \times 1}$, where $z_{i,j}$ is defined in Equation (2)

The next operations are all based on $\tilde{\mathcal{G}}$, where we follow Huang et al. (2020) for the propagation procedure optimization. The initial embedding $\tilde{\mathbf{Z}}_{\text{Logit-LP}}^{(0)}$ is first processed by the sigmoid($\cdot$) function, then the diffusion source embedding $\mathbf{G}$ is set up in the following way. *(1) On the training set of $\mathcal{E}^+/\mathcal{E}^-$*: the node values are 0/1-valued labels minus the initial embedding, referring to the residual error. *(2) On validation and testing sets*: the node values are all-zero embeddings. After Equation (4) generates the final residual errors $\tilde{\mathbf{Z}}^{(k)}|_{k=K_{\max}+1}$, we add the initial values $\tilde{\mathbf{Z}}^0$ and convert the residual to the final result.

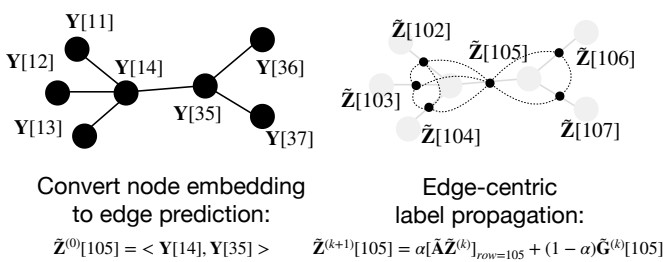

Figure 3: The edge-centric label propagation algorithm demonstrated with a subgraph. The edge prediction is first computed from the node embeddings (GNN output), then the graph is switched to an edge-centric view. The diffusion propagates the residual error from the labeled edges to the entire graph.

Besides the Logit-LP, we also propose and discuss two other ways to shift the original graph to an edge-centric view, namely, the embedding diffusion LP and the XMC diffusion-based LP.

**Variant Model: The Embedding Diffusion-Based LP.** The embedding diffusion LP (Emb-LP) is similar to Logit-LP in that it also performs diffusion on an edge-centric graph $\tilde{\mathcal{G}}$, the nodes of which represent edges in the original graph. The number of nodes of the edge-centric graph, in this case, is $\tilde{\mathcal{G}}$ is $|\mathcal{E}^+|$, which is the number of positive edges in $\mathcal{G}$. In Emb-LP, the initial embedding $\tilde{\mathbf{Z}}^{(0)}$ and the diffusion source embedding $\tilde{\mathbf{G}}$ are identical, which is processed from the GNN output embeddings $\mathbf{Y}$. Denote the embedding for node $i$ and $j$ as $\mathbf{Y}[i]$ and $\mathbf{Y}[j]$. If there is an edge between the node pair $(i,j)$, then in $\tilde{\mathcal{G}}$, the embedding for this edge is the concatenation of $\mathbf{Y}[i]$ and $\mathbf{Y}[j]$. The label propagation procedures are the same as Equation (4). After the propagation, an edge's existence is predicted via the dot product of the original embeddings of its two end nodes.

**Variant Model: The XMC Diffusion-Based LP.** The third LP variant is based on the eXtreme Multi-label Classification (XMC) (Liu et al., 2017; Bhatia et al., 2015) formulation of link prediction (abbreviated as XMC-LP). In the multi-label classification formulation of the link prediction, each node can independently belong to $N = |\mathcal{G}|$ classes (not mutually exclusive). Each class means the existence of the link to the corresponding node (one of $N$ total nodes). XMC-LP operates on the original graph $\mathcal{G}$, and the adjacency matrix is the same as the original one. The initial embedding $\tilde{\mathbf{Z}}^{(0)}$ is set to be the post-dot-product logits and has the shape of $N \times N$. The value at the location $(i,j)$ corresponds to the dot product of the GNN output $\mathbf{Y}[i]$ and $\mathbf{Y}[j]$. The remaining steps are the same as Logit-LP. After performing diffusion using Equation (4), the edge existence between node $i$ and $j$ can be predicted by looking at the location $(i,j)$ of the diffusion result $\tilde{\mathbf{Z}}^{(k)}|_{k=K_{\max}+1}$.

**Summary of Three LP Variants.** The three different views leverage different advantages that LP offers. Logit-LP is supervised by edge labels (positive or negative edges) and performs the best. Emb-LP is unsupervised and is the only variant that outputs embeddings instead of logits. XMC-LP operates on the smaller original graph instead of the edge-centric graph, though it requires more memory.

Our edge-centric LP algorithm is conceptually simple, lightweight, and generic. However, many real-world graphs follow long-tail distributions, with the majority of their nodes having few connections. Some nodes are even isolated, with no connected neighborhood. These cases are shown to have negative effects on message-passing-based approaches (Zheng et al., 2021). Therefore, we also study and compare an MLP-based graph predictor (Hu et al., 2021; Zhang et al., 2021c; Zheng et al., 2021), which has been shown to be effective in sparsely linked graphs.

### 2.4 Alternative Broadcasting Approach for Stage II: Teacher-Student Learning via GNN-MLP

In this alternative approach, we train an MLP with a simple procedure of knowledge distillation from a GNN teacher. Here, the GNN teacher is the same link prediction GNN model discussed previously. The student MLP is first trained to mimic the output of the teacher GNN:

$$\Theta_{\mathrm{MLP}}, \mathbf{X}' = \underset{\Theta_{\mathrm{MLP}}, \mathbf{X}'}{\arg \min} ||\mathbf{Y} - \mathrm{MLP}([\mathbf{X}, \mathbf{X}']; \Theta_{\mathrm{MLP}})||^2 \tag{5}$$

where $\Theta_{\mathrm{MLP}}$ denotes the MLP parameters. After training with this imitation loss until convergence, the MLP is then fine-tuned alone on the link prediction task using the loss in Equation (3). The distilled MLP has no graph dependency during inference, so it can be applied on low-degree nodes. Furthermore, it can generalize well due to the structural knowledge learned from the GNN teacher on the well-connected nodes.

**Comparing Logit-LP and GNN-MLP in Stage II.** So far, we have introduced two options for Stage II broadcasting: a novel edge-centric LP variant and one that adopts off-the-shelf tools (originally developed for accelerated inference and cold-start generalization) in a new context (graph transfer learning). Because this paper focuses on diving into the new GITL setting and workflow, we purposely keep the individual stages simple. The novelty of our method does not lie in inventing Stage II building blocks.

We also clarify another important question: *why do we need two options?* In short, we propose Logit-LP to address the over-smoothness when transferring across source/target graph samples, and conveniently scale up to industry-scale graphs (Chen et al., 2020; Sun & Wu, 2021; Wu et al., 2019; Huang et al., 2020; Chen et al., 2021b; Duan et al.). On the other hand, we develop and study the GNN-MLP as an alternative strategy with complementary merits to tackle the inherently diverse real-world graph learning challenges.

## 3 Empirical Evaluation

### 3.1 Experimental Settings

In this section, we evaluate the proposed framework on several concrete applications and datasets. We use an e-commerce graph dataset of queries and items, and two public benchmarks of social network and citation graph (OGB-collab and OGB-citation2). The statistics of these datasets are summarized in Table 1.

| Datasets | E-commerce (E1) | | OGB-citation2 | | OGB-collab | |
|---|---|---|---|---|---|---|
| Intersection nodes | 1,061,674 | | 347,795 | | 55,423 | |
| Union nodes | 11,202,981 | | 2,927,963 | | 235,868 | |
| **Subgraphs** | **Source** | **Target** | **Source** | **Target** | **Source** | **Target** |
| Num. of Nodes | 10,456,209 | 1,934,188 | 2,604,211 | 671,547 | 218,738 | 77,137 |
| Num. of Edges | 82,604,051 | 6,576,239 | 24,582,568 | 2,525,272 | 2,213,952 | 622,468 |
| Mean Degree | 7.9 | 3.4 | 9.4 | 3.8 | 10.1 | 8.0 |
| Median Degree | 2 | 1 | 6 | 1 | 5 | 5 |

Table 1: The statistics of datasets selected for evaluation.

**Datasets.** The e-commerce dataset is sampled from anonymized logs of a global e-commerce store. The data used here is not representative of production. The source and target graphs correspond to two locales. The

source graph locale has much more frequent user behavior. The graphs are naturally bipartite: the two types of nodes correspond to products and user query terms. The raw texts of the product titles or user queries are available for each node, and the node features are generated from these texts using Byte-Pair Encoding (Heinzerling & Strube, 2018).

OGB-collab (representing a collaboration between co-authors) and OGB-citation2 (representing papers that cite each other) are open-source academic datasets. The *edges* of OGB-collab contain the year that the two authors collaborated and the *nodes* of OGB-citation2 contain the year the corresponding paper was published. To better simulate our setting, we manually split the data into source and target graphs according to time: the collaborations/papers prior to a given year $y^{(h)}$ are organized into the source graph, while the collaborations/papers after a given year $y^{(l)}$ are organized into the target graph. We set $y^{(l)} < y^{(h)}$ so as to ensure the source and target graphs have overlapping nodes.

**Metrics and Baselines.** We mainly use recall as the evaluation metric to judge the model's ability to recover unseen edges in the sparse target graphs. We adopt PLNLP (Wang et al., 2021) as the link prediction baseline (shown as *GNN* in the tables) for comparison as well as the GNN embedding generator in our approach. On the public datasets, we also compare our methods to SEAL (Zhang & Chen, 2018; Zhang et al., 2021a), Neo-GNN (Yun et al., 2021), unsupervised/self-supervised pretraining methods such as EGI (Zhu et al., 2021a) and DGI (Velickovic et al., 2019), and a few heuristics-based approaches, including Common Neighbors (CN), Adamic Adar (AA) (Adamic & Adar, 2003), and Personalized Page Rank (PPR). In the following, we specify the details and discuss the results of the proprietary e-commerce dataset and the two public datasets.

**Special Dataset Processing.** For each dataset, we have slight differences in how we build the source and target graphs. For the proprietary e-commerce recommendation graph, the source and target graphs naturally come from the two locales. We use additional purchase information to build three different views of the data: **E1**, **E2**, **E3**. In **E1**, there exists an edge between query and product if there is at least one purchase. In **E2**, we threshold the number of purchases to be at least three to form the *less connected graph*. This leads to a sparser but cleaner graph to learn from and transfer. **E3** is a graph that uses the human-measured relevance relation as the edges between queries and products. The nodes of **E3** remain the same as **E1**, **E2**, while the edges of **E3** are the edges of **E1** enriched by the positive relevance ratings. Therefore, **E2** is the most sparse graph and **E3** is the most dense one. Table 1 reflects the statistics of **E1**. For **E2**, the mean and median degrees of the source and target graphs are (2.1, 1) and (1.2, 1), respectively. For **E3**, these numbers are (8.3, 2) and (4.2, 1), respectively.

For the open-source graphs, we treat the original graph as the union graph and manually split it into a source graph and a target graph according to the timestamp metadata of the nodes or edges.

## 3.2 Main Results

We next verify the performance of the GITL framework via systematic experiments. [1]. Our results address the following questions.

**Q1: How does the method discover latent edges under different sparsity levels?**

We show the model performances on the e-commerce data in Table 2. In the tables, $N_e^+$ is the number of positive edges in the graph. The *recall @ $N_e^+$* and *recall @1.25$N_e^+$* in the table are the recall numbers for the top $N_e^+/1.25N_e^+$ largest model predictions. To test the model performance with respect to sparsity in the target graph (the key pain point in the GITL setting), we customized three graphs **E1**, **E2**, **E3** with different sparsity levels as specified above, where **E2** is the sparsest one. As can be seen from the results, the proposed Logit-LP performs the best across most sparsity levels.

In a few cases of the less connected edges (**E2**), the propagation is limited due to disconnected subgraphs. In these cases, because the GNN-MLP only requires node features to make predictions, the performance degrades less, making it a better candidate.

---

[1]Our codes are available at `https://github.com/amazon-science/gnn-tail-generalization`

Different models may perform better in different scenarios due to the inherent diversity of real-world data and the difficulty of predicting links under sparse graphs. We suggest simple rules to help practitioners determine which method to use in this special setting. For example, we have found that if the mean degree of the graph is smaller than three, then GNN-MLP may perform better, as shown in the result comparison between E1 and E2, whose mean degrees are 7.9 and 2.1, respectively. Another straightforward guidance is to determine based on their performances on the validation set, which does not add much computational overhead as both GNN-MLP and Logit-LP are lightweight.

| Datasets | Regimes | Tar. $\rightarrow$ Tar. | | Uni. $\rightarrow$ Tar. | | Int. $\rightarrow$ Tar. | |
|---|---|---|---|---|---|---|---|
| | Recall@ | $N_e^+$ | $1.25N_e^+$ | $N_e^+$ | $1.25N_e^+$ | $N_e^+$ | $1.25N_e^+$ |
| | GNN | 86.1 | 89.7 | 73.8 | 77.2 | 90.2 | 93.4 |
| | Emb-LP | 86.8 | 90.0 | 74.0 | 77.6 | 90.6 | 93.7 |
| E1 | Logit-LP | **88.4** | **91.5** | **76.6** | **79.9** | **91.4** | **94.7** |
| | XMC-LP | 86.3 | 89.2 | 73.8 | 77.4 | 90.5 | 93.6 |
| | GNN-MLP | 84.3 | 87.1 | 69.4 | 73.1 | 88.3 | 91.0 |
| | GNN | 84.3 | 86.4 | 71.6 | 74.2 | 86.0 | 89.1 |
| | Emb-LP | 84.7 | 87.1 | 72.4 | 74.9 | 87.2 | 90.3 |
| E2 | Logit-LP | 85.4 | 87.6 | **74.0** | **76.7** | 87.1 | 90.0 |
| | XMC-LP | 83.4 | 85.3 | 72.0 | 73.9 | 86.5 | 89.7 |
| | GNN-MLP | **86.8** | **89.9** | 68.5 | 70.1 | **88.0** | **91.0** |
| | GNN | 66.5 | 68.4 | 62.4 | 65.0 | 69.5 | 72.1 |
| | Emb-LP | 66.9 | 68.8 | **62.9** | **65.7** | 70.1 | 72.6 |
| E3 | Logit-LP | **68.9** | **71.1** | 61.4 | 63.8 | **70.2** | 72.6 |
| | XMC-LP | 66.8 | 68.6 | 62.7 | 65.5 | 69.7 | 72.2 |
| | GNN-MLP | 68.5 | 70.0 | 60.8 | 62.3 | 70.1 | **72.8** |

Table 2: The evaluations for different views of the e-commerce graph. **E1/E2/E3** correspond to the no purchase thresholding (original), purchase thresholded by a minimum of 3, and the user rated relavance indicator graphs. Best results are bolded.

**Q2: How does the proposed framework compared with other methods?**

| Regimes | Tar. $\rightarrow$ Tar. | | Uni. $\rightarrow$ Tar. | | Int. $\rightarrow$ Tar. | |
|---|---|---|---|---|---|---|
| Recall@ | $N_e^+$ | $1.25N_e^+$ | $N_e^+$ | $1.25N_e^+$ | $N_e^+$ | $1.25N_e^+$ |
| GNN (PLNLP) | 51.7 | 62.9 | 49.3 | 60.1 | 51.9 | 63.3 |
| Emb-LP | 52.2 | 63.3 | 49.9 | 61.2 | 52.4 | 63.7 |
| Logit-LP | **55.7** | **65.7** | **51.4** | **63.4** | **55.9** | **65.2** |
| XMC-LP | 52.2 | 63.2 | 49.5 | 61.0 | 52.5 | 63.6 |
| GNN-MLP | 49.8 | 61.4 | 48.5 | 60.0 | 50.6 | 62.9 |
| Neo-GNN | 51.9 | 63.2 | 49.7 | 61.0 | 51.5 | 63.5 |
| CN | 51.1 | 62.5 | 51.1 | 62.5 | 51.1 | 62.5 |
| AA | 50.1 | 62.5 | 50.1 | 62.5 | 50.1 | 62.5 |
| PPR | 51.2 | 62.7 | 50.4 | 61.5 | 51.5 | 63.0 |
| EGI | 52.4 | 64.3 | 50.8 | 61.6 | 53.2 | 65.0 |
| DGI | 52.0 | 64.0 | 50.2 | 61.0 | 52.5 | 64.5 |

Table 3: The recall evaluations on OGB-collab graph.

| Regimes | Tar. $\rightarrow$ Tar. | | Uni. $\rightarrow$ Tar. | | Int. $\rightarrow$ Tar. | |
|---|---|---|---|---|---|---|
| Recall@ | $N_e^+$ | $1.25N_e^+$ | $N_e^+$ | $1.25N_e^+$ | $N_e^+$ | $1.25N_e^+$ |
| GNN (PLNLP) | 46.2 | 58.1 | 47.9 | 60.0 | 48.7 | 60.5 |
| Emb-LP | 45.9 | 58.4 | 48.3 | 60.3 | 48.9 | 60.8 |
| Logit-LP | 47.2 | **61.2** | 48.0 | **63.2** | **51.0** | **64.2** |
| GNN-MLP | 44.0 | 55.9 | 45.2 | 58.1 | 48.2 | 58.8 |
| Neo-GNN | 45.9 | 58.2 | 47.5 | 60.6 | 48.0 | 60.9 |
| CN | 12.2 | 12.5 | 29.7 | 30.3 | 18.9 | 19.4 |
| AA | 12.1 | 12.4 | 29.5 | 30.0 | 18.8 | 19.1 |
| EGI | **48.0** | 60.6 | **49.3** | 62.6 | 49.9 | 62.1 |
| DGI | 47.7 | 59.0 | 49.0 | 60.6 | 49.2 | 61.6 |

Table 4: The recall evaluations on OGB-citation2 graph.

The results of two open-source graphs are shown in Table 3 and Table 4, where the baselines are described in Section 3.1. We see that the proposed edge-centric LP approaches achieve the best performance, better than other state-of-the-art methods in most cases. In contrast, other methods, especially heuristics-based methods, cannot yield meaningful predictions under low degrees (OGB-citation2). To evaluate a pair of nodes, they rely on the shared neighborhoods, which are empty for most node pairs.

| Regimes | Tar. $\to$ Tar. | | Uni. $\to$ Tar. | | Int. $\to$ Tar. | |
|---|---|---|---|---|---|---|
| Recall@ | $N_e^+$ | $1.25N_e^+$ | $N_e^+$ | $1.25N_e^+$ | $N_e^+$ | $1.25N_e^+$ |
| Original Logit-LP | **47.2** | **61.2** | **48.0** | **63.2** | **51.0** | **64.2** |
| Node-centric-LP | 15.3 | 17.6 | 8.5 | 10.4 | 13.5 | 14.7 |
| No $\mathbf{X}'$ Logit-LP | 42.3 | 58.4 | 42.8 | 58.7 | 46.8 | 57.7 |
| No $\mathbf{X}'$ GNN-MLP | 39.6 | 50.1 | 42.3 | 52.9 | 44.1 | 54.0 |
| Logit-LP 50% $\mathcal{E}^-$ | 38.5 | 52.1 | 41.0 | 55.8 | 43.7 | 57.6 |
| Logit-LP 100% $\mathcal{E}^-$ | 45.3 | 58.5 | 46.8 | 61.3 | 49.8 | 62.9 |
| Logit-LP 500% $\mathcal{E}^-$ | 44.8 | 58.0 | 45.2 | 58.9 | 48.1 | 61.8 |
| Logit-LP 1000% $\mathcal{E}^-$ | 39.5 | 54.2 | 41.9 | 54.2 | 44.6 | 58.1 |

Table 5: The ablation studies on OGB-citation2 graph.

| Setting | Precision | | | Accuracy | | |
|---|---|---|---|---|---|---|
| | Uni.$\to$Tar. | Int.$\to$Tar. | Tar.$\to$Tar. | Uni.$\to$Tar. | Int.$\to$Tar. | Tar.$\to$Tar. |
| SEAL | **33.6** | 35.2 | **35.5** | **56.3** | 57.5 | **56.4** |
| Logit-LP | 27.4 | **39.2** | 34.8 | 46.1 | **58.6** | 55.0 |

Table 6: The precision and accuracy metrics on OGB-collab graph.

## Q3: How do the design choices affect the model performance?

We testify several design choices of the proposed method, including the node-centric view of label propagation, the learnable embedding $\mathbf{X}'$, and the ratio of the number of negative edges to the number of positive edges. The results on the citation2 graph are shown in Table 5. As discussed in Section 2.3, the node-centric LP is designed for node-level tasks, hence we switch to an edge-centric view to prevent the over-smoothness of node-centric LP in the link prediction task, as seen in the table. The $\mathbf{X}'$ will be learned in the GNN training stage, and the usage of $\mathbf{X}'$ is to enable more flexible representation space for better performance. The original Logit LP uses 200% negative edges than positive edges. Through the experiments with different numbers of negative edges, we observe that the performance will not be harmed too much if the number of negative edges is not below 100% or more than 500% of positive edges.

## Q4: What are the precision and accuracy metrics?

We present the precision and accuracy results on the OGB-collab dataset in Table 6. As can be seen in Table 6, if we compare across different training set settings, the precision and accuracy numbers are the best when using the intersection-enhanced source graph for training (i.e., the *Int.→Tar.* case). If we compare Logit-LP and SEAL, Logit-LP is better on the *Int.→Tar.* cases while performing worse than SEAL in other cases. Logit-LP performs best for the *Int.→Tar.* case.

## Q5: How does instance selection affect the model performance?

In the tables above, *Int.* $\to$ *Tar.* consistently outperforms other settings. Comparing the *Tar.* $\to$ *Tar.* and *Uni.* $\to$ *Tar.* settings, in the OGB-collab graph, *Uni.* $\to$ *Tar.* is worse, while in the OGB-citation2 graph, the *Uni.* $\to$ *Tar.* is generally better. Nevertheless, we still see that *Int.* $\to$ *Tar.* achieves the best performance among the three settings.

We refer to this phenomenon, where *Int.* $\to$ *Tar.* performs better than *Uni.* $\to$ *Tar.*, as *negative transfer*. This is possibly due to the different distributions of the source graph and the target graph. This lends credence to our hypothesis that adding more data to the source for transfer learning can possibly be counterproductive.

## Q6: Does the model perform better on the source graph compared to the target graph?

We answer this question by comparing the training and test sets, and show the results in Table 7. The training set is mostly composed of the source graph edges while the validation and test sets are only selected from the unshared subgraph of the target graph. From the table, we see that the models behave differently in the source graph (training set) and the target graph (validation/test sets). Since the GNN, LP, and GNN-MLP rely on node features, they perform better on the training set. On the other hand, the featureless heuristics (CN and AA) achieve slightly better performance on the test set.

Comparing the performances on the training and testing sets, most non-heuristic-based methods perform better on the source graph (training set). This performance gap is due to the fact that the edge information is much richer in the source graph. On the other hand, Logit-LP significantly outperforms heuristic-based methods and the GNN, which verifies the effectiveness of the proposed method.

## Q7: How does the intersection size influence the performance?
The influence of intersection size on OGB-collab is shown in Table 8, where we i.i.d. vary the intersection ratio (number of nodes in the shared subgraph, divided by the total node number of target graph). We observe that the performance of

| Regimes | Target → Target | | | Union → Target | | | Intersection → Target | | |
|---|---|---|---|---|---|---|---|---|---|
| Splits | Train | Valid | Test | Train | Valid | Test | Train | Valid | Test |
| GNN (PLNLP) | 64.9 | 37.8 | 37.8 | 55.0 | 39.6 | 39.3 | 62.9 | 39.4 | 39.5 |
| Emb-LP | 65.3 | 38.3 | 38.3 | 55.7 | 40.4 | 39.9 | 63.5 | 40.3 | 40.3 |
| Logit-LP | 65.9 | 38.9 | **38.9** | 56.6 | 41.4 | **40.3** | 64.3 | 41.0 | **41.0** |
| GNN-MLP | 60.9 | 32.9 | 33.6 | 51.5 | 37.4 | 35.9 | 59.8 | 36.6 | 36.2 |
| CN | 10.5 | 11.1 | 11.2 | 26.6 | 27.2 | 27.2 | 16.4 | 17.2 | 17.3 |
| AA | 10.2 | 10.6 | 10.6 | 26.4 | 26.7 | 26.7 | 16.0 | 16.8 | 16.9 |

Table 7: The *recall @* $0.8N_e^+$ expanded for train/validation/test splits of the OGB-citation2 graph.

the proposed method significantly improves as the intersection ratio increases. Even when the ratio is as low as 20%, our proposed transfer performance is higher than the no-transfer baseline. Another natural question is that if the intersection ratio is too small, could the performance be improved by disregarding the intersection structure prior, and using more unshared source graphs as training set. Accordingly, we conducted experiments by adding more nodes in the source graph, so as to contain up to two hops of neighbors of the shared graph. The performances are shown in the "extended" columns. The performances show that extending unshared nodes from the training set does not help much to the model performance. We hypothesize that this may due to the distributional shift between the source and target graphs, which further validates our design choice of leveraging the shared nodes as the structural prior for graph transfer.

| Intersection size | 1% | 1% extended | 5% | 5% extended | 10% | 15% | 20% | 20% extended | 30% | Tar.→Tar. |
|---|---|---|---|---|---|---|---|---|---|---|
| Recall | 13.2 | 12.8 | 32.5 | 34.7 | 53.6 | 55.5 | **56.4** | 54.8 | **56.9** | **55.7** |

Table 8: Influence of the intersection size on OGB-collab. The "extended" means adding two hop neighbors to the training set to mitigate the small training set.

**Q8: How expensive is the computational overhead?** We provide the computational and memory overhead of the LP variants below. For a given graph, denote the number of nodes as $N_{nodes}$, the number of edges as $N_e$, and the number of edges in the edge-centric graph as $N_E$. We can estimate $N_E$ via $N_E \approx 4N_e * \text{mean\_deg}$, where mean_deg is the mean degree of the graph. The number of multiplications during each iteration, the running time, as well as the memory overhead of the LP variants, are summarized in Table 9.

| LP variants | Emb-LP | Logit-LP | XMC-LP |
|---|---|---|---|
| **Num. of multiplications at each iteration** | $N_E * d$ | $N_E * 1$ | $N_e * N_{nodes}$ |
| **Execution time over OGB-citation2** | 1 min 15 s | 2.1 s | 27 min 14 s |
| **Memory overhead after $k$ iterations** | $\mathcal{O}(N_E * d)$ | $\mathcal{O}(N_E * 1)$ | $\mathcal{O}(N_e * k)$ |

Table 9: Computation complexity for different LP variants. The execution time is measured on graph with $N_e = 2,525,272$ edges in the target graph using an 2.90GHz Intel Xeon Gold 6226R CPU.

## 4 Conclusion and Future Work

In this paper, we discuss a special case of graph transfer learning called GITL, where there is a subset of nodes shared between the source and the target graphs. We investigate two alternative approaches to broadcast the link information from the source-enriched intersection subgraph to the full target graph: an edge-centric label propagation and a teacher-student GNN-MLP framework. We demonstrate the effectiveness of our proposed approach through extensive experiments on real-world graphs.

In spirit, this new setting may be reminiscent of previous transfer learning methods, where only a subset of latent components are selectively transferred. Our next steps will study curriculum graph adaptation, in which the target is a composite of multiple graphs without domain labels. This will help us progressively bootstrap generalization across more domains.

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
