# OpenReview forum: "You Only Transfer What You Share: Intersection-Induced Graph Transfer Learning for Link Prediction"
_TMLR — Accepted by TMLR_

### Review · Reviewer_4tmV · 2023-03-06

**Summary Of Contributions:**

This paper identifies a setting of graph transfer learning for link prediction, in which two graphs, one dense (source) and one sparse (target), has a small intersecting subgraph. The subgraph serves as a foundation for knowledge transfer between the source and the target graph that helps the cold-start problem on the target graph.

This paper then proposes two simple but effective techniques for the intersection-based transfer learning setting. First, the authors propose to choose only the intersecting subgraph for pre-training as a technique of source domain sample selection. Second, the authors propose to propagate the logits via a node-edge reversed label propagation (termed the Logit-LP). Finally, the authors propose an alternative to the Logit-LP, namely the GNN-MLP technique, that distills knowledge from a GNN model to an MLP model. In this way, the MLP does not require edges compared to the Logit-LP.

Experiments on three datasets are performed, where the proposed techniques show effectiveness over baselines. Moreover, many analytical experiments are done to analyze various design choices.

**Audience:**

Yes

**Broader Impact Concerns:**

This paper does not raise ethical concerns.

**Claims And Evidence:**

Yes

**Requested Changes:**

I would like to see clarifications on my questions listed in "Weaknesses". My main questions are:
- The random node embeddings $\mathbf{X}'$ cannot adapt to new nodes.
- Can we expand the intersecting subgraph when there are limited intersecting nodes?
- What is a proper size of negative samples required for Logit-LP.

**Strengths And Weaknesses:**

Strengths:
- I appreciate the introduction of the paper. The new setting, Graph intersection-induced transfer learning (GITL) is well motivated and interesting, and the motivation examples are convincing and very clearly stated.
- I also appreciate the overall presentation of this paper. The paper is well positioned over existing works. The technical assumptions are clearly stated. (Section 2.1). The design choices are clearly stated. Overall a very well-written paper.
- I also appreciate that there are two choices, Logit-LP and GNN-MLP to perform link prediction. They well cover different application scenarios.
- Finally, there are extensive experiments. The dataset used cover different sparsity levels and intersection sizes. Ablation studies are quite extensive that compares different source graph samples, different LP strategies, and different intersection sizes.

Weaknesses:

- I have to say that this paper does not have significant weaknesses. I will list some minor points or unclear points for improvements.
- Page 6, the paper says that $\mathcal{E}^* = \mathcal{E}^{src} \cup \mathcal{E}^{tar}$. However, this does not seem accurate, as you say that you only consider edges with one end in $\mathcal{V}^*$.
- Page 6, the paper says that "we concatenate the original node features with a randomly initialized trainable vector $\mathbf{X}'$ for all nodes. However, this operation seems to violate the assumption 1 "there are possibly new nodes being added over time". The randomly initialized $\mathbf{X}'$ cannot generalize to completely new nodes. Also, I would like to see how this operation influences the overall model accuracy, as adding random node embeddings must improve the overall accuracy, but no ablation is done on this design.
- The authors select only the intersection graph (and immediate edges) for source domain instances. From my perspective, the design is more like a tradeoff between number of samples and the closeness between source \& target distributions, instead of a hard choice. For example, in Table 7, the authors show that the performance degrades when a low intersection (<10\%) rate is available. Thus, is it possible to extend the source samples (e.g. two hop neighbors) to complement the lack of intersecting nodes?
- There seems to be an overload of notations in Eqn. 3, as $\Theta, \mathbf{X}'$ exist in both the LHS and the $\arg\max$.
- Also in Eqn. 3, from my understanding, the loss function can also be maximized by pushing $Z[e^+]\rightarrow -\infty$ and $Z[e^-]\rightarrow \infty$, which looks a little strange. Of course, if this loss is widely used and empirically works fine, it is also OK.
- I notice that for the method to work, the negative edge set must be samples at the beginning, and cannot be changed afterwards (as you are going to build nodes based on the negative edge set). This is different from common methods involving negative sampling (e.g. DeepWalk), in which they sample different negative samples at each batch. Therefore, intuitively, the size of $\mathcal{E}^-$ will impact the overall performance, as a very small and fixed negative edge set will cause bias. Maybe an experiment or discussion on it will help clarify.

---

> ### Author Response · Authors · 2023-03-31
> **Author's response to reviewer 4tmV**
>
>
>
> ## Dear reviewer 4tmV:
>
> Thank you for taking the time to review our paper and for providing valuable feedback. We have carefully considered all of your comments and have made the necessary changes to improve the clarity and correctness of our paper. Below we discuss them point by point.
>
> ## $\mathcal{E^*}$ Notation
> For the inaccurate notation of edge union, we appreciate your attention to detail in pointing this out. We have removed the inaccurate notation and used text to explain more clearly in the updated version.
>
> ## Compatibility of $\bf{X’}$ on new nodes
> We acknowledge your concern regarding the randomly initialized embeddings and their ability to generalize to completely new nodes. We have added an ablation study for the random node embedding and put the results in Table 5. The results show that adding random embedding improves the performance, potentially due to the feature-independent embeddings enabling a larger space. Additionally, we note that these feature-independent randomly initialized embeddings can be made compatible with completely new nodes: we can fall back to the first stage to fine-tune the GNN on these new nodes, and then apply Logit-LP after these new nodes are trained with GNN. During the GNN fine tuning stage, the new random embeddings can be optimized. It only fails when the new nodes have not established any link.
>
> ## Extending intersection graph
> For your concern on the possibility to extend the intersection with unshared source graph nodes for low intersection ratio, we have conducted new experiments to determine this. The results are in the updated Table 8. From the results, we did not see significant performance improvement, potentially due to the difference in the distribution. Whether the multi-hop extension will contribute to the performance is up to the data, and it may require dataset-dependent and/or task specific optimizations to improve the performance. Given this observation, we highlight the design choice of using intersection as the most straightforward and simple way to leverage the structural prior, as discussed in the introduction section.
>
> ## Overloaded notations
> Regarding the overloaded notations for $\bf{\Theta}$ and $\bf{X’}$, we’ve simplified the expression and have made it clearer in the updated version.
>
> ## Loss function
> Regarding the loss function for $Z[e^+]$ and $Z[e^-]$, we apologize for the typo. The loss function is to be minimized, not maximized. We have changed it in the updated version. We checked that our code implementations are correct.
>
>
> ## Effect of $\mathcal{E}^-$ size
> Regarding your concern that $\mathcal{E}^-$ cannot be changed, we appreciate your carefulness in pointing this out. We have added an ablation study of varying negative edge sizes in the updated Table 5. The new experiments show that the performance will not be harmed much if the negative edge set is between 100%~500% of the positive edges. If we control the negative sample size about 2 times to the positive edge set (our original setting), the method works reliably.
>
> Thank you once again for your time and valuable feedback. We believe that our updated paper is much improved, and we hope that you will find it satisfactory.
>
>
> Sincerely,
>
> Authors

---

> > ### Comment · Reviewer_4tmV · 2023-04-02
> > **Response acknowledged.**
> >
> > I appreciate the authors for their response. I checked the revised paper and was glad to see that my questions are answered.
> >
> > Cheers.

---

> > > ### Author Response · Authors · 2023-04-02
> > > **Thank you**
> > >
> > > Dear reviewer 4tmV:
> > >
> > > We appreciate your feedback and your acknowledgement.
> > >
> > > Sincerely,
> > >
> > > Authors

---

### Review · Reviewer_rdsk · 2023-03-18

**Summary Of Contributions:**

In this paper, the authors propose an instance-level graph transfer learning framework, called Graph Intersection-induced Transfer Learning (GITL), to alleviate the sparsity issue in link prediction tasks. Specifically, they leverage the intersection subgraph, constructed by the shared nodes between the source graph and target graph, to transfer rich link information from the source graph to the target sparse graph. The learning process consists of two stages. Firstly, they select a set of shared nodes as the training set. Secondly, two approaches are designed to broadcast the initial predictions to the entire target graph. The work is interesting. However, a number of concerns should be addressed.

**Audience:**

Yes

**Claims And Evidence:**

Yes

**Requested Changes:**

Please try to address the mentioned weakness above.

**Strengths And Weaknesses:**

Strengths:

1.The idea to leverage the intersection graph to transfer knowledge is easy to understand, and the approach to shift the role of nodes into edges to avoid the over-smoothness induced by the node-level diffusion is interesting.

2.The writing is clear and the content is readily comprehensible.

Weakness:

1.The manner of label propagation makes the generalization capability of the proposed method rather limited. The method not only requires an intersection between the source graph and the target graph but joint labels, which is not applicable in real-world transfer learning tasks.

2.The contributions of the work should be highlighted. Personally, the reviewer thinks the proposed method cannot be regarded as a graph transfer learning method. Because the method deals with a scenario where an intersection lies between the source graph and the target graph, which contradicts the requirements of transfer learning. Besides, the authors use label propagation to transfer knowledge. However, label propagation is classic semi-supervised learning to deal with i.i.d. data instead of non-i.i.d. data of transfer learning.

3.The authors claimed the proposed method can address the over-smoothness issue. However, no theoretical proof or experiment result can verify the effectiveness of the method. The authors are encouraged to conduct experiments on deeper GNNs and visualize the classification boundaries to verify the effectiveness in addressing the over-smoothness issue.

4.The authors claimed the proposed method is lightweight. Although computation complexity has been analyzed, the training time and the inference time are also should be measured.

5.Equation 3 in section 2.3 models the training objective of the GNN for link prediction as maximizing the sum at the right-hand side. However, as described by [1], the common AUC link prediction loss is minimized.

6.The notation of the intersection graph is ambiguous. In section 2.2, the authors denote the common nodes in two graphs as $\mathcal{V}^* = \mathcal{V}^{src} \bigcap \mathcal{V}^{tar}$, and define the intersection graph as all edges from the source and target graphs that have one end node in $\mathcal{V}^*$, denoted as $\mathcal{E}^ = \mathcal{E}^{src} \bigcup \mathcal{E}^{tar}$. However, the notion $\mathcal{E}^*$ is a little confusing and likely to be mistaken for the union of all edges from the source and target graph.

[1] Wang Z, Zhou Y, Hong L, et al. Pairwise learning for neural link prediction[J]. arXiv preprint arXiv:2112.02936, 2021.

---

> ### Author Response · Authors · 2023-03-31
> **Author's response to reviewer rdsk**
>
>
> ## Dear reviewer rdsk,
>
> Thank you for taking the time to review our paper and providing your feedback. We appreciate your constructive criticism, and have carefully addressed your comments in the weakness section point by point below.
>
> ## Points 1, 2 Problem setting
> Regarding your weakness points 1 and 2, you stated that the "generalization capability of the proposed method is limited because of requiring intersection between the two graphs" and that "the setting is not a standard graph transfer learning". We would like to clarify that it is **our problem specification** that requires source-target graph intersection and joint labels, not **our method**. Our problem setting is abstracted from the day-to-day repeatedly recurring tasks that the author team needs to solve during our operations. We can not overrule its requirements, otherwise, it is solving a different problem, other than the one that we are interested in. This problem setting may not be a hot topic in the graph learning and transfer learning communities, but it is practically important for our organization, and we see it supported by a few real-world applications from both global e-commerce stores and social network scenarios discussed in the introduction section.
> Though our setting does require an intersection subgraph, our method can still work even when the intersection graph is very small. As seen in Table 8 (updated version), our method using a 10%~15% intersection ratio can almost match the performance of the no transfer baseline.
>
>
> We understand that our setting may not be properly referred to as "transfer learning" because the target graph data is not completely out-of-distribution. We hence used a new name, GITL (graph intersection induced transfer learning), and used this new term throughout the paper. If you still think the term "transfer" should be less tied to our setting, we are glad to change the name to "Graph Intersection Induced link prediction boosting" or something similar, and we will add more careful discussions to separate it from transfer learning in a new version.
>
> ## Point 3 Over-smoothness
> Regarding your point 3, you required "experiments on deeper GNNs for over-smooth". We appreciate your advice for strengthening our paper. However, we would like to mention that the visualization of over-smoothness shall better be conducted via comparing the design choice of node-centric view and edge-centric view, instead of comparing against deeper GNNs. As mentioned on page 6, "To avoid the over-smoothness induced by the node-level diffusion of traditional LP, we shift the role of nodes into edges". The way our method avoids over-smoothing is by shifting from traditional node-centric label propagation to edge-centric view, based on our experimental observation that the latter performs better. We have added a comparison of node and edge-centric view methods on two datasets in updated Table 5. We have also compared the result of a deeper GNN for your reference, please check the table below.
>
> ## Point 4 Running time
> Regarding your point 4, we appreciate your advice to use execution time to better support the model’s *lightweight* property. We have now measured the running time and have added these results to the updated Table 9 of the paper.
>
> ## Points 5, 6 Notations
> For point 5, the AUC loss is to be minimized, we are sorry for the typo in the paper. We make sure the implementation is correct. We’ve changed it in the updated version. For point 6, we appreciate the advice that the notation of edge union may be confusing. We’ve removed the notation, and changed it to text description.
>
>
> Thank you again for your valuable feedback. We hope our response helps to resolve the concerns of this paper.
>
> Best regards,
>
> Authors
>
>
>
>
> | Regimes | Tar. $\to$ Tar. |   | Uni. $\to$ Tar. |  | Int. $\to$ Tar. |   |
> |---------|------|----------|-------|----------|--------|---------|
> | Recall@ | $N^+_{e}$ | $1.25N^+_{e}$ | $N^+_{e}$ | $1.25N^+_{e}$ | $N^+_{e}$ | $1.25N^+_{e}$ |
> | GNN (PLNLP) | 46.2 | 58.1 | 47.9 | 60.0 | 48.7 | 60.5 |
> | Logit-LP | 47.2 | 61.2 | 48.0 | 63.2 | 51.0 | 64.2 |
> | Node-centric-LP | 15.3 | 17.6 | 8.5 | 10.4 | 13.5 | 14.7 |
> | GNN-32 layers | 25.7 | 28.4 | 23.4  | 25.3 | 29.4 | 32.5 |
> | GNN-32 layers + Logit-LP | 29.2 | 33.6 | 26.3 | 29.2 | 34.1 | 38.8 |

---

> ### Author Response · Authors · 2023-04-04
> **Sincerely expecting further discussions**
>
> Dear Reviewer rdsk,
>
> We appreciate your time and effort in reviewing our paper. We have addressed all of your concerns in our response and have made the necessary updates to the paper.
>
> As the author response period is ending soon, we would like to know if you require any additional information or clarification based on our reply. Thank you very much for your time and consideration.
>
> Best regards,
>
> Authors

---

> > ### Comment · Reviewer_rdsk · 2023-04-10
> > **Response acknowledged**
> >
> > Thanks for addressing my concerns. I agree with the authors to add more discussions to separate it from transfer learning in a new version and change the name to a new one.

---

### Review · Reviewer_xCMb · 2023-03-21

**Summary Of Contributions:**

This paper proposes Graph Intersection-induced Transfer Learning (GITL), which leverages the information of the shared nodes between the source graph and the target graph for domain transferring. In detail, GITL first trained only the shared nodes and then broadcasts link information from this subgraph to the full target graph. The experimental results seem to validate the effectiveness of the proposed method in domain transferring.

**Audience:**

Yes

**Broader Impact Concerns:**

None.

**Claims And Evidence:**

Yes

**Requested Changes:**

Please see the weaknesses above.

**Strengths And Weaknesses:**

Strengths:

1. The method is well-motivated. Leveraging the intersection between the source graph and the target graph is rational.

2. The authors provide comprehensive experimental results to support their claim.

Weaknesses:

1. The performance gain is inconsistent among the proposed methods. For example, Logit-LP achieves almost the best performance on E1 datasets while GNN-MLP achieves the best performance on E2 datasets. It would be better for the authors to provide guidance on when and where to use the proposed methods.

2. The proposed is only applicable to the case where there are shared nodes between the source graph and the target graph. It would be better for the authors to discuss this limitation in the paper and show whether the proposed method can work when the intersection size is 0%.

---

> ### Author Response · Authors · 2023-03-31
> **Author's response to reviewer xCMb**
>
>
> ## Dear reviewer xCMb:
>
> Thank you for your valuable feedback on our paper. We appreciate the time and effort you put into reviewing our work.
>
> Our paper aims to address the concrete problems which the author team members *repeatedly encounter* during our operations. In these day-to-day occurring problems, there are usually two graphs from two data sources with the same data types who have shared nodes, and we need to optimize link prediction in one graph. Some motivating examples are explained in the introduction section.
>
> Regarding your concern of *inconsistent performance of the proposed methods*, we agree that different models may perform better in different scenarios due to the inherent diversity of real-world data and the difficulty of predicting links under sparse graphs. We do suggest simple rules to help practitioners determine which method to use in this special setting. For example, we have found that if the mean degree of the graph is smaller than three, then GNN-MLP may perform better, as shown in the result comparison between E1 and E2, whose mean degrees are 7.9 and 2.1, respectively. Another straightforward guidance is to determine based on their performances on the validation set, which does not add much computational overhead as both GNN-MLP and Logit-LP are lightweight. We have added these discussions in section 3.2 Q1.
>
> Regarding your comment on the applicability of our method when the intersection size is 0%, we understand your concern. However, this paper is not motivated by this scenario. Before writing this paper, our author team was working on optimizing a set of similar problems that repeatedly occur during our operations. In these problems, we always want to optimize the link prediction in a graph where we have a subset of nodes shared by another source graph. This paper is then summarized based on these concrete and practically important problems. To solve these problems, we want to develop a framework that *leverages the structure prior of shared nodes in the first place*, which we found missing in previous research. Therefore, the case of 0% intersection size is not the problem that we are interested in.
> Though our setting and method do not allow for 0% intersection ratio, our method can still work even when the intersection graph is very small. As seen in Table 8 (updated version), our method using a 10%~15% intersection ratio can almost match the performance of the no transfer baseline.
>
>
> Thank you again for your thoughtful feedback, and we hope our response has addressed your concerns.
>
> Sincerely,
>
> Authors

---

> ### Author Response · Authors · 2023-04-04
> **Sincerely expecting further discussions**
>
> Dear Reviewer xCMb,
>
> We hope this message finds you well. We wanted to check in to see if you had a chance to review our response to your comments on our paper. We have addressed all your concerns and updated the manuscript accordingly.
>
> The author response period is ending soon. Please let us know if you have any additional questions or concerns. We appreciate your time and effort in reviewing our work.
>
> Thank you for your consideration.
>
> Best regards,
>
> Authors

---

### Decision · Action_Editors · 2023-05-08

**Recommendation:** Accept as is

**Comment:**

This paper proposes an instance-level graph transfer learning framework, called Graph Intersection-induced Transfer Learning (GITL), which alleviates the sparsity issue in link prediction tasks. Specifically, GITL leverages the intersection subgraph, constructed by the shared nodes between the source graph and target graph, to transfer rich link information from the source graph to the target sparse graph. The learning process consists of two stages. Firstly, GITL selects a set of shared nodes as the training set. Secondly, two approaches are designed to broadcast the initial predictions to the entire target graph. In general, the main idea of this paper in this paper are interesting. The authors address reviewers' concerns well in their rebuttal. Thus, I would like to recommend accept as is.

**Audience:**

Yes

**Claims And Evidence:**

Yes